# Facial Sadness Recognition is Modulated by Estrogen Receptor Gene Polymorphisms in Healthy Females

**DOI:** 10.3390/brainsci8120219

**Published:** 2018-12-07

**Authors:** Mayra Gutiérrez-Muñoz, Martha E. Fajardo-Araujo, Erika G. González-Pérez, Victor E. Aguirre-Arzola, Silvia Solís-Ortiz

**Affiliations:** 1Facultad de Psicología, Universidad Autónoma de Nuevo León, Monterrey, Nuevo León 66451, Mexico; mlgm.lcp@gmail.com; 2Departamento de Ciencias Médicas, Universidad de Guanajuato, León, Guanajuato 37320, Mexico; mfajardo@ugto.mx (M.E.F.-A.); eg.gonzalezperez@ugto.mx (E.G.G.-P.); 3Facultad de Agronomía, Universidad Autónoma de Nuevo León, Monterrey, Nuevo León 66451, Mexico; victor.aguirrearz@uanl.edu.mx

**Keywords:** menopause, *ESR1*, *ESR2*, emotion, recognition, sadness

## Abstract

Polymorphisms of the estrogen receptor *ESR1* and *ESR2* genes have been linked with cognitive deficits and affective disorders. The effects of these genetic variants on emotional processing in females with low estrogen levels are not well known. The aim was to explore the impact of the *ESR1* and *ESR2* genes on the responses to the facial emotion recognition task in females. Postmenopausal healthy female volunteers were genotyped for the polymorphisms Xbal and PvuII of *ESR1* and the polymorphism rs1256030 of *ESR2*. The effect of these polymorphisms on the response to the facial emotion recognition of the emotions happiness, sadness, disgust, anger, surprise, and fear was analyzed. Females carrying the P allele of the PvuII polymorphism or the X allele of the Xbal polymorphism of *ESR1* easily recognized facial expressions of sadness that were more difficult for the women carrying the p allele or the x allele. They displayed higher accuracy, fast response time, more correct responses, and fewer omissions to complete the task, with a large effect size. Women carrying the *ESR2* C allele of *ESR2* showed a faster response time for recognizing facial expressions of anger. These findings link *ESR1* and *ESR2* polymorphisms in facial emotion recognition of negative emotions.

## 1. Introduction

Facial emotion recognition is essential for efficient social interactions [1]. Facial expressions are a group of emotions that includes happiness, sadness, anger, fear, disgust, and surprise [2]. These emotions have been widely examined to describe the brain areas that modulate them [3], as well as the detection of deficits in the recognition of the facial expressions of emotion in aging [4]. The amygdala is part of the neural circuitry crucial for emotion [5,6] and is involved in fear [7], sadness, and happiness [8], whereas surprised faces are associated with greater activity in the right postcentral gyrus and left posterior insula [9]. The insula is also activated by a broad range of disgust-related stimuli, such as facial expressions of disgust [10,11]. Some studies have shown evidence for sex differences in the identification and processing of emotional facial expressions, including the recognition of emotion [12]. Females tend to display higher accuracy in the identification of facial expression than men [13,14]. Women are also more prone to suffer from affective temperament dysregulation and suicidal behaviors [15,16], which are considered psychopathological diseases that impact the quality of life [17]. Genetic factors have been implicated in the manifestation of these disorders, although the evidence is unclear. It has been found that sex differences influence the manifestation of depression and related disorders, such as anxiety, conduct disorder, and suicidality, and might alter the effects of genetic polymorphisms of the serotonergic system [18]. The monoamine oxidase-A linked polymorphic region (MAO-A), which alters the degradation serotonin and other amine neurotransmitters, has been related with risk for depression and anxiety disorders in postmenopausal women [19]. However, a study did not find an association between MAO-A3 gene variants and increased suicidal risk in patients with chronic migraine and affective temperamental dysregulation measured with specific questionnaires [20].

In addition, there is evidence that sex hormones influence emotional processing [21,22]. For instance, it has been observed that females in the periovulatory phase, which is distinguished by elevated levels of estrogen, were significantly more accurate in recognizing fearful faces than females in the follicular phase of the menstrual cycle with lower estrogen levels [23]. One study showed that, in young women, the percentage of errors was significantly higher for the emotional facial expressions of sadness and disgust in the follicular phase than in the menstrual phase [24]. Sadness was more accurately recognized during the early follicular phase, which is distinguished by lower levels of estrogen, than during the luteal phase of the menstrual cycle [25]. A negative association was reported between facial disgust recognition and estradiol levels, and higher progesterone levels were related with slowing responses to the expression of anger, sadness, happiness, and neutral expressions in naturally cycling females [26]. 

These studies highlight the role of ovarian hormones, particularly estrogens, in the emotional processing of faces [22], but the role of estrogen receptors in the recognition of emotional expressions is rarely studied, particularly during postmenopause. In women, this stage is distinguished by low levels of estrogen and progesterone, producing significant physiological effects [27,28] that have been associated with several adverse outcomes, such as depressive symptoms [29,30], anxiety and severe quality-of-life impairment [31], and complaints of changes in cognitive functions [32]. The estrogen receptor genes are potential genes that might impact the facial emotion recognition due to their relations with cognition [33] and emotion [34]. The estrogen receptor (ER) is a ligand-intensify protein that is a partner of the steroid nuclear receptor family [35,36,37]. Estrogen actions are mediated across estrogen receptors, ER𝛼, ER𝛽, which are fully dispensed over the brain [38], and the G protein-coupled estrogen receptor (GPER) mainly identified in striatum, hypothalamus, hippocampus, and substantia nigra [39]. ER is encoded by two genes, ER𝛼 and ER𝛽, that function both as signal transducers and transcription factors to modulate expression of target-gene [35,40]. The ER𝛼 gene (*ESR1*) is located on chromosome 6q25.1 [41], and the ER𝛽 (*ESR2*) gene is located on chromosome 14q22–24 [42]. ER𝛼 is mainly expressed in the hypothalamus and amygdala [43,44], areas related to autonomic function, emotional regulation [34], associative learning, and attention [45]. ER𝛽 is mainly expressed in the hippocampal and entorhinal cortex [46], brain areas related to declarative memory [47,48]. Two of the most studied polymorphisms in the *ESR1* gene are PvuII (rs9340799) and Xbal (rs223493) [34,49], which are located in the intron of *ESR1* and are in strong linkage disequilibrium with each other [50], which may impact tissue-specific gene expression [51]. 

Several studies have linked the receptor estrogens alpha and beta with cognitive impairment [34] in diverse populations. The X and P alleles of *ESR1* have been associated with elevated risk for cognitive impairment in Alzheimer patients and older adults [33,52,53,54,55]. The X allele of XbaI polymorphism showed an elevated risk in executive cognitive ability, indicating cognitive impairment in postmenopausal women [56]. The GG genotype of rs9340799 was associated with better immediate recall among African and Caucasian women and rs2234693 CC was associated with better performance on the delayed recall task [57]. Older women with a p allele had a greater score decline in Mini-Mental State Examination, while women with at least one x allele also had greater score decline [58]. Another study reported that two of the *ESR1* SNPs (rs8179176, rs9340799) and two of the *ESR2* SNPs (rs1256065, rs1256030) were associated with the likelihood of older women developing cognitive impairment [59]. A recent study found that the XbaI-351 AA genotype was more common amongst subjects with cognitive decline, while -351G allele carriers showed cognitive stability or improvement [60]. Another study reported that the A allele of rs1256049 of *ESR2* polymorphisms was associated with an increased risk of substantial decline in visual memory, psychomotor speed and on the incidence of Mild Cognitive Impairment [61]. Individuals carrying the minor allele of rs7450824 had a lower risk of Alzheimer disease than homozygous subjects for the major allele [62].

Hence, there is evidence of the influence of estrogen receptors alpha and beta on mood and cognition [34], but their relationship with the processing of emotional faces is not well studied. The goal of the present investigation was to explore the impact of estrogen receptor 𝛼 and 𝛽 gene polymorphisms on the response to facial emotion recognition tasks that require the identification of emotional expressions by postmenopausal healthy women with low estrogen levels. It was hypothesized that the genetic variants PvuII, Xbal, and rs1256030 of estrogen receptor alpha and estrogen receptor beta confer susceptibility to recognize facial expressions of surprise, anger, happiness, disgust, sadness, and fear emotions.

## 2. Materials and Methods

### 2.1. Participants

Sixty-nine postmenopausal healthy female volunteers between 48 and 60 years old were genotyped for the Xbal, PvuII, and rs1256030 polymorphisms of estrogen receptor genes in a crossover design. The sample size of 69 women with the Xbal polymorphism was calculated to yield an expected power of 0.82 to detect a difference of 10% on a facial emotion recognition task with a two-sided significance level of α = 0.05 [63]. The following inclusion criteria were considered: this study assessed a group of females in the postmenopausal period, which is distinguished by a decline of estrogens [28], because it has been found that variations in ovarian hormones modulate emotional processing [22]. In addition, only females were also chosen because they tend to distinguish facial emotions with more accuracy than men [64]. The females intervened a medical examination to evaluate their health condition. To take part in the investigation, females had been amenorrheic for at least 12 months, with no record of cardiovascular, diabetes, metabolic, endocrine or cancer and none of them was on any kind of medication or had ever taken hormones [63]. Initial dementia was excluded employing the Mini-Mental State Examination (MMSE), points of this exam range from 0 to 30, and people with dementia usually score under 24 [65]. In the current investigation, the MMSE scores of the participants ranged from 27 to 30. Major depression was rejected via the Beck Depression Inventory [66], and only females scored between 0 and 9 were incorporated in the study. Each female was evaluated in a unique session by one qualified female researcher during the same time of day (between 09:00 h and 11:00 h) [63]. In the present study, the visual acuity of the participants was not evaluated by an objectively measured visual acuity to determine visual problems such as glaucoma, macular degeneration or other visual disease that could influence the recognition test. Instead of this procedure, those participants with visual problems were asked to wear glasses with their corrected vision. This investigation was accepted by the Institutional Ethics Committee for Research on Human Subjects and is in accordance with the Declaration of Helsinki [67]. All females gave their written informed consent before enrolling in the investigation.

### 2.2. Genotyping 

#### 2.2.1. *ESR1* Gene Polymorphisms 

To detect the *ESR1* gene variants, genomic DNA was extracted from peripheral blood leucocytes using High Pure PCR Template Preparation kit (Roche). For the *ESR1* Xbal and PvuII polymorphisms, the method previously described by Lui et al. [68] was used. Briefly, a genomic fragment of 1430 bp was amplified by PCR with the primers 5′-CTGCCACCCTATCTGTATCTTTTCCTATTCTCC-3′ and 5′-TCTTTCTCTGCCACCCTGGCGTCGATTATCTGA-3′. The PCR conditions were as follows: 30 cycles of denaturation (94 °C, 30 s), annealing (61 °C, 40 s), and extension (72 °C, 90 s). After amplification, the products were digested at 37 °C overnight with either PvuII or Xbal restriction endonucleases, and then electrophoresis was performed in a 1.5% agarose gel stained with ethidium bromide. To identify the genotypes, lowercase letters (p and x for PvuII and Xbal, respectively) were used to indicate the presence of the restriction site for each endonuclease, and uppercase letters (P and X) were used to indicate the absence of the restriction sites. Participants were categorized as PP or XX homozygotes, Pp or Xx heterozygotes and pp or xx homozygotes according to the digestion pattern. 

#### 2.2.2. *ESR2* Gene Polymorphisms

The *ESR2* polymorphism (rs1256030) was genotyped by sequencing. A genomic fragment of 158 bp was amplified by PCR. For the polymorphism rs1256030 with the primers Forward: 5′ GGTAAGATTTGATCTGGCCA 3′ and Reverse: 5′ CTGTGGGGAATGACTAATGTT 3′. The PCR conditions were as follows: 35 cycles of denaturation (94 °C, 30 s), annealing (63 °C, 30 s), and extension (72 °C, 30 s). The PCR products were purified with columns (Wizard PCR Clean Up system, Promega, Madison, WI, USA). Then, amplified DNA fragments were subjected to direct cycle sequence analysis using the Taq dye-deoxy terminator method and an ABI PRISM 3100 Genetic Analyzer (PE Applied Biosystems, Foster City, CA, USA). The genotypes were classified according to the base identified at position 49 of the 158 bp amplicon. Genotype CC showed a single peak emission for the base fluorescence of C. The TT genotype was recorded when a single peak for T was observed, and the CT genotype was identified by the observation of two different peaks. 

### 2.3. Facial Emotion Recognition Task

A computerized task, the Emotion Recognition Task (ERT) [69], was employed to quantify the recognition of six basic facial emotional expressions: anger, disgust, fear, happiness, sadness, and surprise [2]. This task is based on a delayed matching-to-sample paradigm, which requires that a subject learns to give a particular response in the presence of a discriminative or simple stimulus [70,71]. The ERT has been useful to detect variations related to the menstrual cycle and hormones in the processing of emotional faces [24,25,26] and in amygdala activation during an explicit emotion recognition task [72] in young females. The stimuli employed in the ERT were 26 faces from the 110 Pictures of Facial Affect Series [2], randomly selected to consist of 5 facial expressions of happiness, 4 of surprise, 5 of sadness, 4 of fear, 4 of anger, and 4 of disgust made by female and male actors. The pictures were shown on a screen facing the females with the STIM^2^ cognitive system (NeuroScan), with a program designed for the present study, following the procedure previously described by Gasbarri et al. [24]. The STIM^2^ computer registered the numbers of correct responses, errors, and omissions (when the subjects did not give an answer within 3 s) and the response time in ms (time used to answer) for each emotional face from each participant. 

### 2.4. Hormone Quantifications 

Following a procedure previously described by Solis-Ortiz et al. [63], a 10 mL blood sample was extracted from the females. ELISA was employed to quantify 17β-estradiol and progesterone levels. Commercially available radioimmunoassay kits were employed to quantify luteinizing hormone (LH) and follicle stimulating hormone (FSH) levels. The hormone levels were utilized to verify the hormone condition of the postmenopausal females [73,74]. 

### 2.5. Questionnaires

#### 2.5.1. Mood Questionnaire

Depressive symptoms were evaluated through self-report using the Beck Depression Inventory [66] in a standardized version for the Mexican population [75]. This questionnaire consists of 21 items that measure current depressive symptoms. Each item contains a group of four statements, from which the subject chooses one according to how she felt in the last week. Items 1 to 13 assess symptoms that are psychological in nature, while items 14 to 21 assess more physical symptoms. The total score is obtained by adding the scores for the 21 items, with 0 as the lowest score and 64 as the maximum score. 

#### 2.5.2. Mini Mental Test

The cognitive state of participants was evaluated with The Mini-Mental State Examination (MMSE) [65], which is used to evaluate cognitive function among the elderly. The MMSE includes tests of attention, memory, orientation, language, and visual–spatial skills. Scores on this test range from 0 to 30, and subjects with dementia score below 24.

### 2.6. Statistical Analysis

Statistical analysis was executed with STATISTICA for Windows 23 (StatSoft, Inc., Tulsa, OK, USA). Following a procedure previously described by Solis-Ortiz et al. [63], the statistical power of the sample [76] was calculated using the Sample Size Calculation and Power Analysis module in STATISTICA. It was employed α (two-tailed) = 0.05 and β = 0.20 to estimate a standardized effect size of 10% applying standard methods [77]. Data were examined for a normal distribution using the Kolmogorov–Smirnov test before statistical methods were employed. Descriptive statistics were employed to report the features of the females. All measures from ERT scores were converted to z-scores to compare the values between the *ESR1* PvuII P alleles (PP + Pp) and p, Xbal X alleles (XX + Xx) and x, and the *ESR2* C alleles (CC + TC) vs. TT. This statistical tool produces new variables with a standardized value and describes the location of the value in terms of the standard deviation relative to the mean [78]. A separate Mann–Whitney U test and Student’s t-test were computed to compare the *z*-scores from the task between the PvuII P-p, Xbal X-x, and C-T alleles of the *ESR1* and *ESR2* polymorphisms. The Visual Statistics System (ViSta) for Windows 7.9 module “Effect size” was used to correct the data outliers [63] and estimate the effect sizes [79]. Cohen’s d was calculated as the effect size measure and was indicated by the coefficient (d) between allele groups [80]. The coefficient d values were squared to facilitate interpretation in terms of the percentage of the total variance associated with an effect [63]. Differences were considered significant with alpha levels of *p* < 0.05. In the current investigation, it was only examined one specific locus per gene; therefore, it is not required to correct the *p*-value for multiple testing [81]. 

## 3. Results

### 3.1. Features of Females 

Features of the females and the hormonal levels, expressed as the mean and standard deviation, are shown in Table 1. The hormonal levels were within the expected ranges for postmenopausal healthy females [82].

### 3.2. ESR1 Gene Polymorphisms 

#### 3.2.1. PvuII Polymorphism

Genetic analysis identified 11 postmenopausal females with the PP genotype (15.94%), 32 with the Pp genotype (46.38%), and 26 with the pp genotype (37.68%), a distribution consistent with Hardy–Weinberg equilibrium (*X*^2^ = 0.048, *p =* 0.82). To examine the connection between *ESR1* polymorphisms and the recognition emotional faces, groups of PP + Pp genotype carriers were joined into a group denominated the P allele group to facilitate comparisons with the pp allele group due to the little size of females and to raise the statistical power.

The results of the facial ERT for surprise, anger, happiness, disgust, sadness, and fear scores were transformed from task variables into z-scores for the allele groups P and p for the PvuII polymorphism and are shown in Figure 1. As shown in panel a, the response times to recognize facial expressions of surprise (*t =* −0.648, *p =* 0.51), anger (*t =* −0.480, *p =* 0.63), happiness (*t =* −1.259, *p =* 0.21), disgust (*t =* −0.527, *p =* 0.59), and fear (*t =* −0.734, *p =* 0.46) were not significantly different between the two allele groups. The response time to recognize sadness was significantly different between the two allele groups (*t =* −2.620, *p =* 0.01, d = 0.090, *p =* 0.01, explaining 9.0% of the entire variance in the data). The z-score demonstrated that females carrying the *ESR1* P allele had a faster response time for recognizing facial expressions of sadness, as indicated by a mean response time (M = 0.45, SD = 0.26) that was shorter than that of the p allele group (M = 1.59, SD = 0.06), which had a longer mean response time. Panel b shows that the recognition accuracy for sadness was significantly different between the two alleles (*t =* 3.01, *p =* 0.003, d = 0.109, *p =* 0.006, explaining 10.9% of the total variance in the data) and had a large effect size. The z-score indicated that females carrying the *ESR1* P allele recognized facial expressions of sadness with better accuracy, as indicated by an accuracy percentage (M = 0.59, SD = 0.03) that was markedly higher than the mean for the *ESR1* p allele group (M = −1.53, SD = 0.08). The recognition accuracy for surprise (*t =* −0.118, *p =* 0.90), anger (*t =* 0.421, *p =* 0.67), happiness (*t =* 0.120, *p =* 0.90), disgust (*t =* −0.501, *p =* 0.61), and fear (*t =* 0.416, *p =* 0.67) did not differ significantly between the two allele groups. As shown in panel c, the correct responses to recognize surprise (*t =* −0.118, *p =* 0.90), anger (*t =* 0.421, *p =* 0.67), happiness (*t =* 0.120, *p =* 0.90), disgust (*t =* −0.501, *p =* 0.61), and fear (*t =* 0.416, *p =* 0.67) were not significantly different between the two allele groups. The numbers of correct responses in recognizing sadness were significantly different between the two alleles (*t =* 3.016, *p =* 0.003, d = 0.109, *p =* 0.006, explaining 10.9% of the total variance in the data). The z-score demonstrated that females carrying the *ESR1* P allele produced more correct responses for recognizing facial expressions of sadness, as indicated by a larger mean number of correct responses (M = 1.10, SD = 0.05) than in the *ESR1* p allele group (M = 0.01, SD = 0.02). As shown in panel d, the numbers of omissions in response to surprise (*t =* −0.155, *p =* 0.87), anger (*t =* −1.585, *p =* 0.11), happiness (*t =* −1.175, *p =* 0.24), disgust (*t =* 1.233, *p =* 0.22), and fear (*t =* −0.726, *p =* 0.47) were not significantly different between the two allele groups. The numbers of omissions in response to faces showing sadness were significantly different between the two allele groups (*t =* −3.393, *p =* 0.001, d = 0.136, *p =* 0.002, explaining 13.6% of the total variance in the data). The z-score demonstrated that females carrying the *ESR1* P allele omitted fewer responses when shown facial expressions of sadness, as indicated by a lower mean number of omissions (M = −0.14, SD = 0.02) than for the *ESR1* p allele group (M = 1.70, SD = 0.06). The errors committed in recognizing facial expressions of surprise (U = 555, *p =* 0.93), anger (U = 461, *p =* 0.14), happiness (U = 520, *p =* 0.49), disgust (U = 545.5, *p =* 0.90), fear (U = 532, *p =* 0.73), and sadness (U = 508, *p =* 0.38) were not considerably distinctive between the two allele groups.

#### 3.2.2. Xbal Polymorphism

Genetic analysis identified 5 females with the XX (7.35%) genotype, 34 with the Xx genotype (50.00%), and 29 with the xx genotype (42.65%), a distribution consistent with Hardy–Weinberg equilibrium (*X*^2^ = 1.37, *p =* 0.24). To examine the connection between *ESR1* polymorphisms and the recognition of emotional faces, groups of XX + Xx genotype carriers were joined into a group denominated the X allele group to facilitate comparisons with the xx allele group due to the little size of females and to raise statistical power.

The results of the facial emotion recognition task, given as the scores for surprise, anger, happiness, disgust, sadness, and fear from the variable task, were transformed to z-scores for the two allele groups *ESR1* X and *ESR1* x for the Xbal polymorphism and are shown in Figure 2. As shown in panel a, the response times for recognizing surprise (U = 553, *p =* 0.88), anger (U = 493.5, *p =* 0.37), happiness (U = 506.5, *p =* 0.46), disgust (U = 537, *p =* 0.73), and fear (U = 514, *p =* 0.53) were not significantly different between the two allele groups. The response time for recognizing sadness was significantly different between the two alleles (U = 439, *p =* 0.05, d = 0.043, *p =* 0.08, explaining 4.30% of the total variance in the data). The *z*-score demonstrated that women carrying the *ESR1* X allele had a faster response time for facial expressions of sadness, as indicated by a mean response time (M = 0.33, SD = 0.05) shorter than that for the *ESR1* x allele group (M = 1.61, SD = 0.10). In panel b, the recognition accuracy for surprise (U = 508.5, *p =* 0.39), anger (U = 440.5, *p =* 0.08), happiness (U = 498.5, *p =* 0.30), and disgust (U = 455.0, *p =* 0.12) were not significantly different between the two allele groups. However, the recognition accuracy for sadness was significantly different between the two allele groups (U = 414.5, *p =* 0.03, d = 0.053, *p =* 0.05, explaining 5.3% of the total variance in the data). The z-score demonstrated that females carrying the *ESR1* X allele recognized the facial expressions of sadness more accurately, as indicated by a mean accuracy (M = 0.48, SD = 0.02) higher than that of the *ESR1* x allele group (M = −1.88, SD = 0.15). The recognition accuracy for fear was also significantly different between the two alleles (U = 405.5, *p =* 0.03, d = 0.057, *p =* 0.05, explaining 5.7% of the total variance in the data). The z-score demonstrated that women carrying the *ESR1* X allele recognized the facial expressions of fear with lower accuracy, as indicated by a mean accuracy (M = −1.01, SD = 0.10) lower than that of the *ESR1* x allele group (M = 0.21, SD = 0.02). As shown in panel c, the numbers of correct responses for recognizing surprise (U = 508.5, *p =* 0.39), anger (U = 440.5, *p =* 0.09), happiness (U = 498.5, *p =* 0.30) and disgust (U = 455, *p =* 0.12) were not significantly different between the two allele groups. The correct responses for recognizing sadness were significantly different between the two alleles (U = 414.5, *p =* 0.03, d = 0.053, *p =* 0.05, explaining 5.3% of the total variance in the data). The z-score demonstrated that women carrying the *ESR1* X allele were better able to recognize facial expressions of sadness, as indicated by a higher mean number of correct responses (M = 0.99, SD = 0.07) than in the *ESR1* x allele group (M = 0.20, SD = 0.01). The numbers of correct responses for recognizing fear were also significantly different between the two alleles (U = 405.5, *p =* 0.03, d = 0.057, *p =* 0.05, explaining 5.7% of the total variance in the data). The z-score demonstrated that women carrying the *ESR1* X allele had fewer correct responses for recognizing facial expressions of fear, as indicated by a lower mean number of correct responses (M = −0.81, SD = 0.04) than in the *ESR1* x allele group (M = −0.55, SD = 0.02). As shown in panel d, the omissions when recognizing the emotions surprise (U = 549, *p =* 0.79), anger (U = 516, *p =* 0.49), happiness (U = 565, *p =* 0.99), and fear (U = 536.5, *p =* 0.64) were not significantly different between the two allele groups. The omissions in response to disgust were significantly different between the two allele groups (U = 445.5, *p =* 0.05, d = 0.068, *p =* 0.03, explaining 6.8% of the total variance in the data). The z-score demonstrated that women carrying the *ESR1* X allele omitted more responses to facial expressions of sadness, as indicated by a higher mean number of omissions (M = 0.27, SD = 0.01) than that for the *ESR1* x allele group (M = −0.82, SD = 0.05). The numbers of omissions when recognizing sadness were also significantly different between the two alleles (U = 386.5, *p =* 0.006, d = 0.095, *p =* 0.01, explaining 9.5% of the total variance in the data). The *z*-score demonstrated that women carrying the *ESR1* X allele omitted fewer responses to facial expressions of sadness, as indicated by a lower mean number of omissions (M = −0.38, SD = 0.01) than in the *ESR1* x allele group (M = 1.88, SD = 0.10). There were no significant differences between the two allele groups in the numbers of errors made during recognition of facial expressions of surprise (U = 519.5, *p =* 0.39), anger (U = 472, *p =* 0.15), happiness (U = 486.5, *p =* 0.16), disgust (U = 544.5, *p =* 0.71), fear (U = 467, *p =* 0.12) and sadness (U = 564, *p =* 0.98).

### 3.3. ESR2 rs1256030 Polymorphism 

Genetic analysis identified 17 females with the CC (27.42%) genotype, 33 with the CT genotype (53.23%), and 12 with the TT genotype (19.35%), a distribution consistent with Hardy–Weinberg equilibrium (*X*^2^ = 3.16, *p =* 0.57). To examine the connection between *ESR2* polymorphisms and the recognition of emotional faces, groups of CC+CT genotype carriers were joined in a group denominated the C allele group to facilitate comparisons with the TT allele group due to the little size of females and to raise the statistical power.

The results of the facial emotion recognition task for surprise, anger, happiness, disgust, sadness, and fear scores were transformed from task variables into z-scores for the two allele groups, *ESR2* C and *ESR2* T, for the *ESR2* (rs1256030) polymorphism and are shown in Table 2. The response time to recognize facial expressions of surprise (U = 161.5, d = 0.067, *p =* 0.03, explaining 6.7% of the total variance in the data) and anger (U = 155.0, d = 0.093, *p =* 0.02, explaining 9.3% of the total variance in the data) were significantly different between the two alleles. The z-score demonstrated that women carrying the *ESR2* T allele showed slower response times for recognizing facial expressions of surprise (M = −1.61, SD = 0.08), as indicated by a shorter-than-average response time. Women carrying the *ESR2* C allele showed a faster response time for recognizing facial expressions of anger (M = 1.37, SD = 0.05), as indicated by a longer-than-average response time. There were no significant differences between the two allele groups regarding accuracy, correct responses, omission, and errors when recognizing facial expressions of surprise, anger, happiness, disgust, sadness, and fear.

## 4. Discussion

The findings of the current investigation demonstrated significant effects of the genetic variants PvuII and Xbal of the polymorphic genes *ESR1* and *ESR2* on facial emotion recognition ability in postmenopausal females. Our outcome displayed that the recognition of facial expression of sadness is modulated by *ESR1* gene polymorphisms. Females carrying the *ESR1* P allele of the PvuII polymorphism recognized facial expressions of sadness more easily than the women carrying the *ESR1* p allele, as indicated by their higher accuracy, faster response time, greater numbers of correct responses, and fewer omissions on the facial emotional recognition task. This result explained 10.9%, 9.0%, 10.9%, and 13.6% of the entire variance, respectively, and it also showed a large effect size. Similarly, women carrying the *ESR1* X allele of the Xbal polymorphism were better able to recognize the facial expressions of sadness than the women carrying the *ESR1* x allele, as indicated by the higher accuracy, fast response time, more response corrects, and fewer omissions committed on the facial emotional recognition task. This outcome explained 5.3%, 4.3%, 5.3%, and 6.8% of the total variance, respectively, and it also showed a medium effect size. Interestingly, no effect of the *ESR1* gene polymorphisms was found on recognition of the facial expressions of surprise, anger, and happiness. The *ESR2* only showed a significant effect in the response time to recognize facial expressions of surprise and anger.

The identification of facial expressions of sadness by postmenopausal females carrying *ESR1* and *ESR2* gene polymorphisms found in the present study suggests an effect of the expression of estrogen receptors on emotion processing. However, previous studies reported in the literature relating to estrogen receptors and emotion recognition are scarce. One study reported that females carrying rare alleles of the three *ESR2* SNPs, rs928554, rs1271572, and rs1256030, which are in linkage disequilibrium with each other, displayed superior face recognition compared with noncarriers in a face memory task, and suggested that estrogen receptors may regulate social memory functions in humans [1]. Coincidentally, our results based on the facial emotional recognition task found that postmenopausal females carrying the *ESR2* C allele showed a faster response time for recognizing facial expressions of anger, as indicated by a response time shorter than the mean for females carrying the *ESR2* T allele. In addition, there is evidence that *ESR2* are linked to anxiety disorders and can act as modulators of anxiety-like behavior [83]. An investigation noted that, for the ERβ gene *ESR2*, the SNP rs1256049 was related with a greater incidence of generalized anxiety disorder in older women [84], while the female ERβ knockout mice increased anxiety-like behaviors [85]. A study analyzed the anxious behavior resulting from the deletion of genes involved in anxiety to their brain expression in mice, with one of them being the *ESR2* gene [86]. This study found that the genes accompanied, after deletion, by a modification of the anxious behavior, have lower expression in the cerebral cortex, the amygdala and the ventral striatum, brain regions involved in the emotional circuit, and that presynaptic genes are involved in the emergence of anxiety and postsynaptic genes in the reduction of anxiety after gene deletion. [86]. In addition, it has been described that ESR1 are also involved in the motivational system [87], and that a hypertrophy of mesocorticolimbic system is associated with increased activity and impaired attention of Naples high excitability rats [88]. 

Some studies conducted in young females have linked the facial expression of emotion with variations in sex hormones during the menstrual cycle [22]. Although these studies did not analyze estrogen receptors, they support the results found in the present study. Better emotion recognition accuracy for sad faces has been reported in the early follicular phase, which is distinguished by low levels of estrogen [25]. Another study reported working memory deficits for sad and disgusted faces in the early follicular phase, which is related to low estrogen and progesterone levels, compared with the luteal phase of the menstrual cycle in healthy women [24]. Better emotion recognition accuracy was reported in the early follicular phase in healthy young women [89]. These outcomes are partially consistent with our results because postmenopause is also distinguished by low levels of serum estrogens. Moreover, our research found better facial sadness recognition related to certain estrogen receptor *ESR1* gene polymorphisms. It is not known how this variant could influence estrogen signaling and to affect the recognition of sad expressions. It is likely that the Xbal and PvuII SNPs as well as the TA-repeat do not affect estrogen signaling but may be in linkage disequilibrium with other yet-unidentified causative functional variant in the *ESR1* gene [34,50,90]. 

The ERT employed in the current investigation requires the use of working memory, which implies to the mechanisms that are considered to be essential in order to keep things in mind while executing complex tasks such as reasoning, understanding, and learning [91]. Working memory is modulated by the prefrontal cortex [92], where estrogen receptors have been found [93], indicating that estrogen may affect prefrontal functions [94]. Furthermore, estrogens increase the levels of neurotransmitters, promote neuronal growth and synapse formation, and regulate second messenger systems [95,96]. Thus, a decrease in estrogen, which occurs during postmenopause, could be detrimental to working memory processes in prefrontal function [97]. A previous study reported that executive function was lower in late postmenopause than the women in the early postmenopausal stage, while there were no changes in other cognitive domains [97]. Associative memory and episodic verbal memory performance were worse in postmenopausal women, suggesting deficits in executive functions [98,99]. In the current investigation, it was remarkable to note that postmenopausal females carrying the P allele of the PvuII polymorphism or the X allele of the Xbal polymorphism of *ESR1* displayed higher accuracy, fast response time, more correct responses, and fewer omissions to complete the task, with a large effect size. These results suggest that ER alpha maintains transcription and memory as estradiol levels decline [100], facilitating the execution of the task in healthy females. 

It is also significant to remark that the postmenopausal females examined in the current investigation were in good health without any evident record of diagnosis of major depression or cognitive alterations, which were detected by self-report questionnaires. Nevertheless, the results of the current investigation demonstrate that females carrying the P allele of PvuII and the X allele of Xbal, two polymorphisms of *ESR1*, displayed better accuracy in the recognition of sad expressions, which could reflect a subtle propensity for depression during the postmenopausal period. Some studies have considered the possibility that vision impairment is related to symptoms of anxiety and depression in the elderly [101], although the present study did not measure the visual acuity of postmenopausal females with a vision test. Evidence suggests that including a measurement for visual impairment may be relevant. For instance, one study found that in a sample of young and middle-aged U.S. adults with worse visual function displayed increased odds of having an anxiety disorder [102]. Another study which carried out a systematic review found that depression was associated with visual impairment, although the studies considered in the review lacked a standardized evaluation of visual acuity, which makes it difficult to establish a reliable association with the depressive symptoms [103]. Thus, future studies should consider measuring visual impairments in emotional processing. 

Several studies have suggested that *ESR1* has a role in depression [34,104]. These receptors are mainly expressed in the hypothalamus and amygdala [43,44,105], areas related in autonomic function and emotional regulation [34,106]. Genetic variation in the estrogen receptors may modify estrogen signaling, thus influencing a woman’s susceptibility to developing depression [107]. A study conducted on Chinese women diagnosed with major depression and healthy controls found that women carrying the genotype PP of the PvuII SNP had a higher risk of major depression compared to women with the Pp or pp genotype [108]. Women homozygous for the variant X allele of *ESR1* rs9340799 had an increased risk of diagnosis for major depression across their lifetime compared with women who were homozygous for the x allele, while the XX genotype of rs9340799 was specifically associated with an increased risk of recurrent depressive episodes [104]. Women carrying the AA alleles in the *ESR1* polymorphism rs9340799 had an increased lifetime prevalence of major depression among postmenopausal female users of hormonal therapy [109]. Another study conducted in postmenopausal females found that the *ESR1* 454(-351) AG and 454(-351) AG + GG genotypes were related with lower risk of depression [110]. Moreover, it was also reported that females with the AA alleles of *ESR2* rs4986938 had a higher lifetime prevalence of major depression than women with GA and GG genotypes [109]. 

The molecular mechanisms by which these polymorphisms alter the activity of the receptor are not clear, since PvuII and Xbal are located in an intronic and apparently nonfunctional region of the gene [50]. Potential reasons comprehend the existence of functional combinations among polymorphic alleles wherein two markers in blend would modify the genetic function as well as the stability of the RNA [111]. Another reason would be that polymorphisms in the intron could be in linkage disequilibrium with the exon, which would affect the function of the estrogen receptor [112]. 

Moreover, there is evidence linking variations in estrogen levels with susceptibility to depression, particularly in females during postmenopause [113,114,115], which is distinguished by low serum estrogens [28], consistent with the findings of the current study. Estrogen modulates serotonergic function [116,117,118,119], which has been correlated to several aspects of mood and emotional information processing [120]. Some studies that employed a facial emotion recognition task showed that patients with a diagnosis of major depression tend to be better at identifying negative stimuli, such as sad faces [121], and to show increased vigilance and selective attention towards sad expressions [122], that women with depression in remission correctly identified more emotions as anger, fear, and sadness than controls [123] and that patients with depression had a higher accuracy rate for recognition of sadness compared to controls [124]. 

The administration of hormone replacement therapy has been used to examine the effects during emotional processing in females. For example, a study revealed that postmenopausal women treated with estrogens showed higher activation than women without estrogens treatment in the entorhinal cortex during processing of negative pictures, whereas women who used estrogens recently showed greater activation in the hippocampus and higher emotion recognition accuracy of neutral stimuli [125]. Another study reported that postmenopausal women treated with estrogen plus progesterone showed activation in the orbital, frontal, cingulate, and occipital cortices during processing of negative emotional pictures [126]. Moreover, it was observed that the administration of raloxifene, a selective estrogen receptor modulator, binds estrogen receptor alpha (ESR-α) [127], increased PFC activity during inhibition of response to negative words and was greater in patients with schizophrenia homozygous for *ESR1* rs9340799 AA relative to G carriers [128].

There are some limitations that must be considered in the present study. This study was conducted in a specific sample of Mexican postmenopausal women with low estrogen levels. The study did not include females with high estrogen levels, which must be included in future approaches to determine whether there are differences associated with the estrogen receptors. It also must include the analysis of other polymorphisms associated with estrogen and serotonin metabolisms. Other studies must include individuals of different ages and gender to consider sex differences in the mood and cognition.

The findings of the present study have some implications for the clinical practice. Women in early postmenopause may be prone to depression, because they are carriers of a variant of the estrogen receptor that confers risk to depression. This could have important preventive and therapeutic implications, to develop specific medicines based on the genetic profile of people with depression, especially in those people who do not respond to conventional treatment. 

## 5. Conclusions 

The actual outcomes add novel details about the impact of estrogen receptors on facial emotion recognition in postmenopausal healthy females with reduced estrogen levels. The outcomes of the current investigation show better recognition of the facial expression of sadness by carriers of the P allele and the X allele of *ESR1* and *ESR2* gene polymorphisms, but no such differences were found for facial expressions of happiness. These findings suggest that emotional processing displays contrasts depending on the estrogen receptor gene variants, which in part describes individual contrasts in the identification of facial aspect of sadness manifested by postmenopausal healthy females. 

## Figures and Tables

**Figure 1 brainsci-08-00219-f001:**
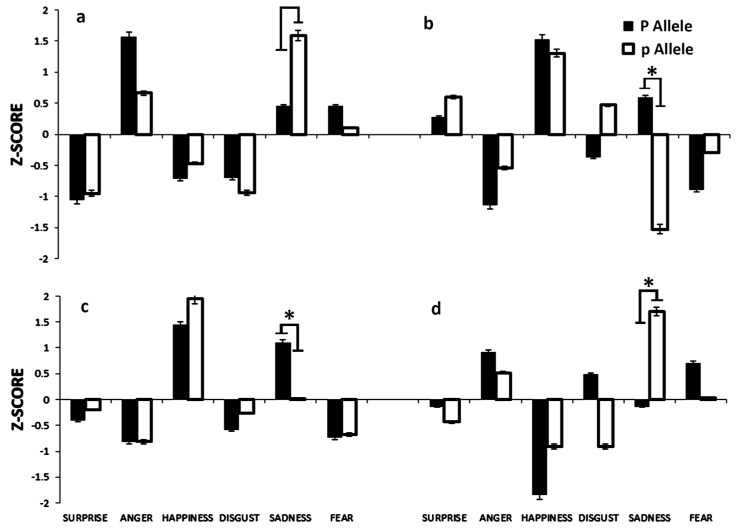
Shows the *z*-scores of the facial emotion recognition task for the two allele groups *ESR1* P and p of the PvuII polymorphism. The panels show response time (**a**), accuracy (**b**), correct responses (**c**), and omissions (**d**). Asterisks indicate significant differences between the allele groups (* *p* < 0.05).

**Figure 2 brainsci-08-00219-f002:**
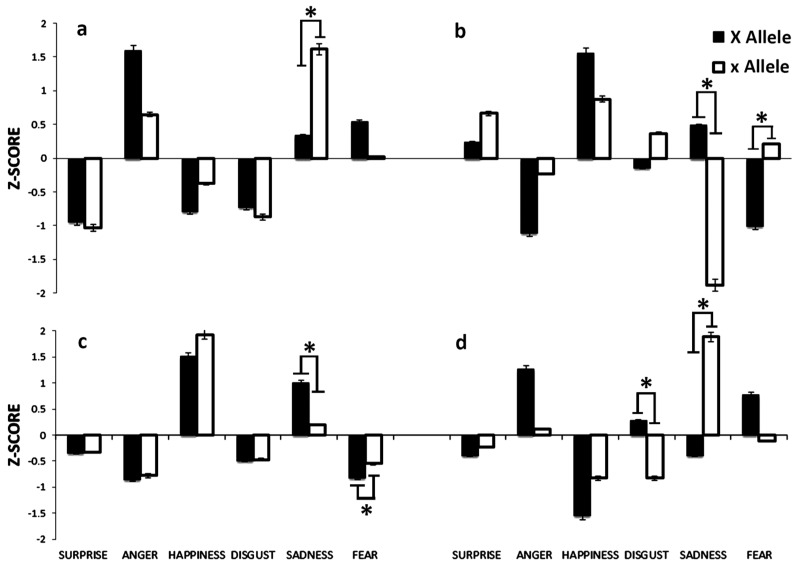
Shows the *z*-scores of the facial emotion recognition task for the two allele groups *ESR1* X and x of the Xbal polymorphism. The panels show response time (**a**), accuracy (**b**), correct responses (**c**), and omissions (**d**). Asterisks indicate significant differences between the allele groups (* *p* < 0.05).

**Table 1 brainsci-08-00219-t001:** Women characteristics.

	(*n* = 69) Mean ± SD
Age (years)	54.47 ± 4.70
Years of education	10.21 ± 3.44
Menarche (years)	12.86 ± 1.44
Menopause (years)	47.03 ± 4.85
Weight (Kg)	67.37 ± 13.20
Height (cm)	1.56 ± 0.06
BMI (Kg/m^2^)	27.83 ± 5.10
TAS (mmHg)	105.90 ± 13.90
TAD (mmHg)	77.34 ± 10.47
FSH (mlU/mL)	64.17 ± 24.90
LH (mlU/mL)	31.42 ± 14.97
Estradiol (ng/mL)	12.45 ± 2.04
Progesterone (ng/mL)	0.55 ± 0.11

BMI = Body Mass Index, TAS = Systolic arterial tension, TAD = Diastolic arterial tension, LH = Luteinizing Hormone, FSH = Stimulating Follicle Hormone.

**Table 2 brainsci-08-00219-t002:** Results of the facial emotion recognition task scores transformed from task variables into *z*-scores for the two allele groups, *ESR2* C and *ESR2* T, for the *ESR2* (rs1256030) polymorphism.

	C Allele Mean ± SD	T Allele Mean ± SD	U Value	Cohen´s d	*p*
**Response Time**					
Surprise	−1.15 ± 0.08	−1.61 ± 0.09	161.5	0.067	0.03 *
Anger	1.37 ± 0.05	0.24 ± 0.01	155.0	0.093	0.02 *
Happiness	−0.47 ± 0.03	−0.49 ± 0.02	186.5	0.046	0.11
Disgust	−0.86 ± 0.04	−0.10 ± 0.01	248.0	0.005	0.69
Sadness	0.85 ± 0.04	1.24 ± 0.05	255.0	0.005	0.79
Fear	0.25 ± 0.02	0.73 ± 0.03	244.0	0.005	0.63
**Accuracy**					
Surprise	0.77 ± 0.03	1.01 ± 0.03	262.5	0.002	0.99
Anger	−1.39 ± 0.05	−0.30 ± 0.01	209.5	0.023	0.23
Happiness	1.32 ± 0.05	1.40 ± 0.03	251.5	0.001	0.72
Disgust	0.34 ± 0.02	−0.96 ± 0.03	261.0	0.005	0.88
Sadness	−0.41 ± 0.02	−0.17 ± 0.01	257.0	0.004	0.79
Fear	−0.63 ± 0.02	−0.96 ± 0.03	261.5	0.001	0.85
**Correct Responses**					
Surprise	−0.27 ± 0.04	−0.29 ± 0.01	262.5	0.002	0.99
Anger	−0.99 ± 0.10	−0.62 ± 0.05	209.5	0.023	0.23
Happiness	1.57 ± 0.15	1.49 ± 0.15	251.5	0.001	0.72
Disgust	−0.41 ± 0.05	−0.79 ± 0.05	261.0	0.005	0.88
Sadness	0.85 ± 0.08	1.00 ± 0.07	257.0	0.004	0.79
Fear	−0.74 ± 0.08	−0.79 ± 0.05	261.5	0.001	0.85
**Errors**					
Surprise	−1.62 ± 0.10	−1.11 ± 0.05	266.0	0.001	0.93
Anger	1.03 ± 0.05	1.11 ± 0.04	258.5	0.001	1.00
Happiness	−0.29 ± 0.02	1.11 ± 0.03	257.5	0.005	0.86
Disgust	0.14 ± 0.01	−1.11 ± 0.03	239.0	0.013	0.45
Sadness	−0.29 ± 0.02	0.18 ± 0.01	253.5	0.001	0.68
Fear	1.03 ± 0.04	0.18 ± 0.01	261.5	0.002	0.89
**Omissions**					
Surprise	−0.63 ± 0.05	−0.45 ± 0.02	266.5	0.001	1.00
Anger	1.07 ± 0.10	−0.45 ± 0.02	209.0	0.034	0.20
Happiness	−1.24 ± 0.10	−1.54 ± 0.13	236.5	0.025	0.34
Disgust	−0.63 ± 0.05	1.18 ± 0.07	220.5	0.027	0.19
Sadness	1.20 ± 0.10	0.63 ± 0.03	252.0	0.006	0.70
Fear	0.22 ± 0.02	0.63 ± 0.03	245.5	0.001	0.64

* *p* < 0.05, Mann–Whitney U Test.

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
