# Peer review of "Facial Sadness Recognition is Modulated by Estrogen Receptor Gene Polymorphisms in Healthy Females"

_brainsci, 2018, doi:10.3390/brainsci8120219_

Reviewer 1 Report

The manuscript entitled "Facial Sadness Recognition is Modulated by Estrogen Receptor Gene Polymorphisms in Healthy Females" by Gutiérrez-Muñoz et al describes the different performance of postmenopausal woman in the recognition of facial expressions based on the polymorphism of ESR.
The mauscript is interesting, the language is fluent, aims are clear and methods appropriate.
I have only few comments.
1) please verify if there is any change in visual acuity,  any change in mood, which might modify the performance in facial expression perception. If this is not possible, please discuss this item in the discussion section.
2) Please mention that ESR2 gene is related to anxiety and that a recent analysis suggests its developmental role in brain regions related to mood (Viggiano A, Cacciola G, Widmer DAJD, Viggiano D (2015) Anxiety as a neurodevelopmental disorder in a neuronal subpopulation: Evidence from gene expression data. Psychiatry Res 1–12)
3) Please also mention that ESR modifies the motivational system and the hypertrophy of the motivational system is known to be linked to changes in reward perception (Viggiano D, Sadile A (2000) Hypertrophic A10 dopamine neurones in a rat model of attention-deficit hyperactivity disorder (ADHD). Neuroreport 11:3677–3680)

Reviewer 2 Report

This is, in summary, an interesting paper aimed to analyze the impact of the ESR1 and ESR2 genes on the responses to the facial emotion recognition task in females. The authors reported that females carrying the P allele of the PvuII polymorphism or the X allele of the Xbal polymorphism of ESR1 easily recognized facial expressions of sadness that were more difficult for the women carrying the p allele or the x allele. The authors added that they showed higher accuracy, fast response time, more correct responses and fewer omissions to complete the task, with a large effect size. Females carrying the ESR2 C allele of ESR2 showed a faster response time for recognizing facial expressions of anger as well.

The authors may find as follows my main comments/suggestions.

First, when throughout the Introduction section, the authors referred to existing evidence related to the influence of estrogen receptors alpha and beta on both mood and neurcognition, they should more extensively describe, according to the current literature, the hypothesized framework between estrogen receptors alpha and beta on either mood and neurocognition.

In addition, as the authors within the same section correctly focused on sex differences in the identification and processing of emotional facial expressions, including the recognition of emotion and referred to the importance of sex hormones in influencing emotional processing (together with the enhanced likelihood to feel anger and sadness), they could also mention that, when compared to males, females often exhibit affective temperament dysregulation and suicidal behaviors. These specific characteristics may significantly contribute to the psychosocial impairment and altered quality of life of these individuals. Generally, the psychological distress in females may significantly and negatively modify the quality of life and psychosocial functioning. In order to briefly address this topic (although i understand that this is not the main topic of the present manuscript), i suggest to cite and discuss the study published in 2011 on Eur Rev Med Pharmacol Sci (PMID: 23104655).

Moreover, the inclusion/exclusion criteria of the present study need to be better specified for the general readership.

Also, the Mini-Mental State Examination used to exclude subjects with impaired neurocognition and Beck Depression Inventory used to exclude depressed individuals, respectively, need to be, at least briefly, described in the Methods section.

Furthermore,  the assumption that estrogen may affect prefrontal functions is generally interesting for the readers but needs to be better developed.

Importantly, the major limitations/shortcomings of this study need to be more directly reported in order to elucidate the general caveats of this paper for the general readership.

Finally, what is the take-home message of this manuscript? While the authors mentioned that emotional processing displays contrasts depending on the estrogen receptor gene variants, they failed to provide some conclusive remarks to this regard for clinicians. Specifically, based on their expertise, what are the main implications of the present findings in the clinical practice?

Author Response

Reviewer 2

1)In the introduction section, line 86, the following paragraph was added to briefly describe the findings of estrogen receptors on cognition more relevant in humans:

Several studies have linked the receptor estrogens alpha and beta with cognitive impairment [34] in diverse populations. The X and P alleles of ESR1 have been associated with elevated risk for cognitive impairment in Alzheimer patients and older adults [52-56]. The X allele of XbaI polymorphism showed an elevated risk in executive cognitive ability indicating cognitive impairment in postmenopausal women [57].The GG genotype of rs9340799 was associated with better immediate recall among African and Caucasian women and rs2234693 CC was associated with better performance on the delayed recall task [58].Older women with a p allele had a greater score decline in Mini-Mental State Examinationwhile  women with at least one x allele also had greater score decline [59]. Another study reported that two of the ESR1 SNPs (rs8179176, rs9340799) and two of the ESR2 SNPs (rs1256065, rs1256030) were associated with the likelihood of older women developing cognitive impairment  [60]. A recent study found thatthe XbaI- 351 AA genotype was more common amongst subjects with cognitive decline, while -351G allele carriers showed cognitive stability or improvement [61]. Another study reported that the A allele of rs1256049 of ESR2 polymorphisms was associated with an increased risk of substantial decline in visual memory,psychomotor speed and on the incidence of Mild Cognitive Impairment [62].Individuals carrying the minor allele of rs7450824 had a lower risk of Alzheimer disease than homozygous subjects for the major allele [63].

In the discussion section, lines 370-402, we mentioned and discussed the relevant studies on estrogen receptors associated with mood.

In the references section, the following references were added:

52. Isoe, K.; Ji, Y.; Urakami, K.; Adachi, Y.; Nakashima, K. Genetic association of estrogen receptor gene polymorphisms with Alzheimer’s disease. Alzheimer’s Res.1997, 3, 195–197.

53. Brandi, M.L.; Becherini, L.; Gennar, L.; Racchi, M.; Bianchetti, A.; Nacmias, B.; Sorbi, S.; Mecocci, P.; Senin, U.; Govoni, S. Association of the estrogen receptor alpha gene polymorphisms with sporadic Alzheimer’s disease. Biochem. Biophys. Res. Commun.1999, 265, 335–338.

54. Ji, Y.; Urakami, K.; Wada-Isoe, K.; Adachi, Y.; Nakashima, K. Estrogen receptor gene polymorphisms in patients with Alzheimer’s disease, vascular dementia and alcohol-associated dementia. Dement.Geriatr. Cogn. Disord.2000, 11, 119–122.

55. Ryan, J.; Carrière, I.; Carcaillon, L.; Dartigues, J.F.; Auriacombe, S.; Rouaud, O.; Berr, C.; Ritchie, K.; Scarabin, P.Y.; Ancelin, M.L.Estrogen receptor polymorphisms and incident dementia: the prospective 3C study.Alzheimers Dement. 2014, 10, 27-35. doi: 10.1016/j.jalz.2012.12.008.

56. Ma, S.L.; Tang, N.L.; Leung, G.T.; Fung, A.W.; Lam, L.C.Estrogen receptor αpolymorphisms and the risk of cognitive decline: A 2-year follow-up study.Am. J. Geriatr. Psychiatry. 2014, 22, 489-498. doi: 10.1016/j.jagp.2012.08.006.

57. Olsen, L.; Rasmussen, H.B.; Hansen, T.; Bagger, Y.Z.; Tankó, L.B.; Qin, G.; Christiansen, C.; Werge, T. Estrogen receptor alpha and risk for cognitive impairment in postmenopausal women. Psychiatr. Genet.2006, 16, 85–88.

58. Kravitz, H.M.; Meyer, P.M.; Seeman, T.E.; Greendale, G.A.; Sowers, M.R. Cognitive functioning and sex steroid hormone gene polymorphisms in women at midlife. Am. J. Med.2006, 119(9 Suppl 1):S94– S102.

59. Yaffe, K.; Lui, L.Y.; Grady, D.; Stone, K.; Morin, P. Estrogen receptor 1 polymorphisms and risk of cognitive impairment in older women. Biol. Psychiatry. 2002, 51, 677-682.

60. Yaffe, K.; Lindquist, K.; Sen, S.; Cauley, J.; Ferrell, R.; Penninx, B.;Harris, T.; Li, R.; Cummings, S.R. Estrogen receptor genotype and risk of cognitive impairment in elders: findings from the Health ABC study. Neurobiol Aging. 2009, 30, 607-614.

61. Chaves, A.C.; Fraga, V.G.; Guimarães, H.C.; Teixeira, A.L.; Barbosa, M.T.; Carvalho, M.D.; Mota, A.P.; Silva, I.F.; Caramelli, P.; Gomes, K.B.; Alpoim, P.N.Estrogen receptor-alpha gene XbaI A > G polymorphism influences short-term cognitive decline in healthy oldest-old individuals. Arq. Neuropsiquiatr. 2017, 75, 172-175. doi: 10.1590/0004-282X20170018.

62. Ryan, J.; Carrière, I.; Amieva, H.; Rouaud, O.; Berr, C.; Ritchie, K.; Scarabin, P.Y.; Ancelin, M.L.Prospective analysis of the association between estrogen receptor gene variants and the risk of cognitive decline in elderly women. Eur. Neuropsychopharmacol. 2013, 23, 1763-1768. doi: 10.1016/j.euroneuro.2013.06.003.

63. Goumidi, L.; Dahlman-Wright, K.; Tapia-Paez, I.; Matsson, H.; Pasquier, F.; Amouyel.; P.; Kere, J.; Lambert, J.C.; Meirhaeghe, A.Study of estrogen receptor-αand receptor-βgene polymorphisms on Alzheimer's disease. J. Alzheimers. Dis. 2011, 26, 431-439. doi: 10.3233/JAD-2011-110362.

2)In the introduction section, line 42, the following paragraph was added, following the indications of the reviewer:

Women are also more prone to suffer from affective temperament dysregulation and suicidal behaviors  [15,16,], which are considered psychopathological diseases that impact on the quality of life [17]. Genetic factors have been implicated in the manifestation of these disorders, although the evidence is unclear. It has been found that sex differences influence the manifestation of depression and related disorders, such as anxiety, conduct disorder and suicidality, and might alter the effects of genetic polymorphisms of the serotonergic system [18]. The monoamine oxidase-A linked polymorphic region (MAO-A), which alters the degradation serotonin and other amine neurotransmitters, has been related with risk for depression and anxiety disorders in postmenopausal women [19]. However, a study did not find an association between MAO-A3 gene variants and increased suicidal risk in patients with chronic migraine and affective temperamental dysregulation measured with specific questionnaires [20].

In the references section, the following references were added:

15. Sadeh, N.; Javdani, S.; Finy, M.S.; Verona, E. Gender differences in emotional risk for self- and other-directed violence among externalizing adults. J. Consult. Clin. Psychol. 2011, 79, 106-117.

16. Iliceto, P.; Pompili, M.; Lester, D.; Gonda, X.; Niolu, C.; Girardi, N.; Rihmer, Z.; Candilera, G.; Girardi, P. (2011). Relationship between temperament, depression, anxiety, and hopelessness in adolescents: A structural equation model. Depress. Res. Treat. 2011, 160175.

17. Dadomo, H.; Grecucci, A.; Giardini, I.; Ugolini, E.; Carmelita, A.; Panzeri, M. Schema therapy for emotional dysregulation: Theoretical implication and clinical applications. Front. Psychol.2016, 7:1987. doi: 10.3389/fpsyg.2016.01987.

18. Perry, L.M.; Goldstein-Piekarski, A.N.; Williams, L.M.Sex differences modulating serotonergic polymorphisms implicated in the mechanistic pathways of risk for depression and related disorders.J. Neurosci. Res. 2017, 95, 737-762. doi: 10.1002/jnr.23877.

19.Słopień, R.; Słopień, A.; Różycka A.; Warenik-Szymankiewicz, A.; Lianeri, M.; Jagodziński, P.P. The c.1460C>T polymorphism of MAO-A is associated with the risk of depression in postmenopausal women. ScientificWorldJournal 2012, 2012:194845. doi: 10.1100/2012/194845

20. Serafini, G.; Pompili, M.; Innamorati, M.; Gentile, G.; Borro, M.; Lamis, D.A.; Lala, N.; Negro, A.; Simmaco, M.; Girardi, P.; Martelletti, P.Gene variants with suicidal risk in a sample of subjects with chronic migraine and affective temperamental dysregulation. Eur. Rev. Med. Pharmacol. Sci.2012, 16, 1389-1398.

3) In the method section, in line 118, the following sentence was added:

   … The following inclusion criteria were considered:…

4) In the method section, line 188, the following paragraph was added, which briefly describes the questionnaires that were used in the present investigation:

2.5. Questionnaires 

2.5.1. Mood questionnaire

Depressive symptoms were evaluated through self-report using the Beck Depression Inventory [67] in a standardized version for the Mexican population [76]. This questionnaire consists of 21 items that measure current depressive symptoms. Each item contains a group of four statements, from which the subject chooses one according to how she felt in the last week. Items 1 to 13 assess symptoms that are psychological in nature, while items 14 to 21 assess more physical symptoms. The total score is obtained by adding the scores for the 21 items, with 0 as the lowest score and 64 as the maximum score.

2.5.2. Mini Mental Test

The cognitive state of participants was evaluated with The Mini-Mental State Examination (MMSE) [66], which is used to evaluate the cognitive function among the elderly. The MMSE  includes tests of attention, memory, orientation, language and visual-spatial skills.Scores on this test range from 0 to 30, and subjects with dementia score below 24.

In the references section, the following references were added:

76. Jurado, S.; Villegas, M.E.; Méndez, L.; Rodríguez, F.; Loperena, V.; Varela, R. La estandarización del inventario de la depresión de Beck para los residentes de la ciudad de México.Sal. Mental. 1998, 21,36-38.

5) In the discussion section, line  442, the following paragraph was addedto expand knowledge about working memory and estrogens

Furthermore, estrogens increase the levels of neurotransmitters, promote neuronal growth and synapse formation, and regulate second messenger systems [96,97]. Thus, a decrease in estrogen, which occurs during postmenopause, could be detrimental to working memory processes in prefrontal function [98]. A previous study reported that executive function was lower in late postmenopause than the women in the early postmenopausal stage, while there were no changes in other cognitive domains [98].Associative memory and episodic verbal memory performance were worse in postmenopausal women, suggesting deficits in executive functions [99,100].

In the references section, the following references were added:

96. McEwen, B. S.; Milner, T. A. Understanding the broad influence of sex hormones and sex differences in the brain. J. Neurosci. Res.2017, 95, 24-39.doi: 10.1002/jnr.23809

97. McEwen, B. S.; Akama, K. T.; Spencer-Segal, J. L.; Milner, T. A.;Waters, E. M. Estrogen effects on the brain: actions beyond the hypothalamus via novel mechanisms. Behav. Neurosci. 2012, 126, 4-16. doi: 10.1037/a0026708

98, Elsabagh, S.; Hartley, D. E.; File, S. E. Cognitive function in late versus early postmenopausal stage.Maturitas2007, 56, 84-93.

99, McCarrey, A. C.; Resnick, S. M. Postmenopausal hormone therapy and cognition. Horm. Behav.2015, 74, 167–172. doi:  10.1016/j.yhbeh.2015.04.018

100. Rentz, D. M.; Weiss, B. K.; Jacobs, E. G.; Cherkerzian, S.; Klibanski, A.; Remington, A.; Aizley, H.; Goldstein, J. M. Sex differences in episodic memory in early midlife: impact of reproductive aging.  Menopause 2017, 24, 400-408.

6) In the discussion section, line 497, the following paragraph was addedto mention the limitations of the present study:

Limitations

There are some limitations that must be considered in the present study. This study was conducted in a specific sample of Mexican postmenopausal women with low estrogen levels. The study did not include females with high estrogen levels, which must be included in future approaches to determine whether there are differences associated with the estrogen receptors. It also must include the analysis of other polymorphisms associated with estrogen and serotonin metabolisms. Other studies must include individuals of different ages and genderto consider sex differences in the mood and cognition.

7)In the discussion section, line 504, the following paragraph was addedto mention the implications in clinical practice.

The findings of the present study have some implications for the clinical practice. Women in early postmenopause may be prone to depression, because they are carriers of a variant of the estrogen receptor that confers risk to depression. This could have important preventive and therapeutic implications, to develop specific medicines based on the genetic profile of people with depression, especially in those people who do not respond to conventional treatment.

Reviewer 3 Report

Overall, an interesting study that was well written. The introduction could also include studies that examined emotion processing in postmenopausal women who have and have not used hormone therapy (e.g. Shafir et al Behav Brain Res 2012). I wonder whether the discussion could benefit from studies that have looked at estrogen receptor modulators in association with this SNP and emotional response inhibition - that is, to further support the involvements of ESR-1 genotypes in emotion processing (e.g. Kindler et al Eur Neuropsychopharmacol. 2016). Minor point: Table 1 (height should be in m or correct decimal place)

Author Response

Reviewer 3

1)In the discussion section, in line  485, the following paragraph was addedto mention some studies about the influence of hormone replacement therapy on emotional processing:

The administration of hormone replacement therapy has been used to examine the effects during emotional processing in females. For example, a study revealed that postmenopausal women treated with estrogens showed higher activation than women without estrogens treatment in the entorhinal cortex during processing of negative pictures, whereas that women who used estrogens recently showed greater activation in the hippocampus and higher emotion recognition accuracy of neutral stimuli [126].Another study reported that postmenopausal women treated with estrogen plus progesterone showed activation in the orbital, frontal, cingulate, and occipital cortices during processing of negative emotional pictures [127].  

In the references section, the following references were added:

126. Shafir, T.; Love, T.; Berent-Spillson, A.; Persad, C.C.; Wang, H.; Reame, N.K.; Frey, K.A.; Zubieta, J.K.; Smith, Y.R. Postmenopausal hormone use impact on emotion processing circuitry. Behav Brain Res2012, 226, 147-153. doi: 10.1016/j.bbr.2011.09.012.

127. Love, T.; Smith, Y.R.; Persad, C.C.; Tkaczyk, A.; Zubieta, J.K.Short-term hormone treatment modulates emotion response circuitry in postmenopausal women. Fertil. Steril. 2010, 93, 1929-1937. doi: 10.1016/j.fertnstert.2008.12.056.

2) In the discussion section, line 492 , the following paragraph was addedto mention the involvement of ESR1 in emotional response inhibition:

  Moreover, it was observed that the administration of raloxifene, a selective estrogen receptor modulator, binds estrogen receptor alpha (ESR-α) [128] increased PFC activity during inhibition of response to negative words and was greater in patients with schizophreniahomozygous for ESR-1 rs9340799 AA relative to G carriers  [129].

In the references section, the following references were added:

128. Bryant HU. Mechanism of action and preclinical profile of raloxifene, a selective estrogen receptor modulation. Rev.Endocr.Metab.Disord. 2001, 2; 129-38.

129. Kindler, J., Weickert, C.S.; Schofield,P.R.; Lenroot,R.; Weickert,T.W. Raloxifene increases prefrontal activity during emotional inhibition in schizophrenia based on estrogen receptor genotype. Eur.Neuropsychopharmacol. 2016, 26, 1930-1940. doi: 10.1016/j.euroneuro.2016.10.009

3) In Table 1 was corrected decimal place.

Round  2

Reviewer 1 Report

The authors have responded appropriately to my concerns

Reviewer 2 Report

In the revised manuscript, the authors addressed most of the major comments raised by Reviewers improving both the main structure and quality of this paper. I have no further additional comments.